# Assessing COVID-19 Lockdowns' Impacts on Global Urban PM<sub>2.5</sub> Air Quality with Observations and Modeling

Claire M. Yu<sup>1,+</sup>, Mian Chin<sup>1,\*</sup>, Qian Tan<sup>2,3</sup>, Huisheng Bian<sup>1,4</sup>, Peter R. Colarco<sup>1</sup>, Hongbin Yu<sup>1,\*</sup>

<sup>1</sup>Earth Sciences Division, NASA Goddard Space Flight Center, Greenbelt, Maryland, USA

<sup>2</sup>Bay Area Environmental Research Institute, Moffett Field, California, USA

<sup>3</sup>NASA Ames Research Center, Moffett Field, California, USA

<sup>4</sup>GESTAR II, University of Maryland Baltimore County, Baltimore, Maryland, USA

<sup>+</sup> Summer intern at NASA GSFC in 2022 and 2023; now an undergraduate at Duke University

Correspondence to: Hongbin Yu (Hongbin.Yu@nasa.gov), Mian Chin (Mian.Chin@nasa.gov)

Abstract. The regional lockdowns, implemented around the world over 2020-2022 to contain the rapid spread of the novel coronavirus disease 2019 (COVID-19), inadvertently created a natural laboratory for investigating the effect of reducing anthropogenic emissions on urban air quality in unprecedentedly large temporal and spatial scales. In this study, we analyze multi-year surface PM<sub>2.5</sub> observations in 21 cities around the globe to examine anomaly of PM<sub>2.5</sub> (particulate matter with an aerodynamic diameter of less than 2.5 µm) concentrations during major COVID-19 lockdowns with respect to that in the prepandemic years. We then use a set of Goddard Earth Observing System (GEOS) global aerosol transport modeling experiments to disentangle the effect of the lockdown emission reductions from other non-lockdown effects. Our analysis shows that no systematic reductions in PM<sub>2.5</sub> are found in response to the lockdowns globally. In some locations, we find the coincidences of an increasing stringency index and a decreasing of surface PM<sub>2.5</sub>, which often leads to the record low of PM<sub>2.5</sub> over extensive period. These observations clearly suggest the positive impacts of COVID-19 lockdown-induced anthropogenic emission reductions on air quality. In other stations, however, the lockdown's impacts could be masked by differing meteorology and the occurrence of dust and wildfire events. We also found that current satellite remote sensing of aerosol optical depth cannot be used to reliably discern the change of surface PM<sub>2.5</sub> due to the COVID-19 lockdowns. Results of this study provide a preview of potential mixed effects on urban air quality when implementing air pollution control regulations such as transitioning gasoline and diesel-powered vehicles to electric vehicles.

#### 1. Introduction

PM<sub>2.5</sub>, particulate matter (also referred to aerosol in climate communities) with an aerodynamic diameter of smaller than 2.5 μm, is one of the major air pollutants that adversely affect human health. Exposure to PM<sub>2.5</sub> was ranked as the fifth largest contributing factor to global mortality (Cohen et al., 2017). Currently, about 90% of the global population lives in unhealthy environments where annual PM<sub>2.5</sub> concentrations are greater than the guideline of 5 μg m<sup>-3</sup> recently issued by the World Health Organization (WHO) (Yang et al., 2022). Further, it is estimated that the long-term exposure to ambient PM<sub>2.5</sub> might have resulted in the premature deaths of 2.9 (1.4 – 4.5) million in 2019, with more than two thirds occurring in Asia (Yang et al.,

2022). Another study has estimated as many as 8.7 million premature deaths per year that can be attributed to the exposure to PM<sub>2.5</sub> (McDuffe et al., 2021). Clearly, reducing the primary and precursor emissions of PM<sub>2.5</sub> is necessary to improve air quality and the well-being of citizens. Moreover, emissions reductions should focus on anthropogenic sources because they are responsible for most of the mortality attributable to PM<sub>2.5</sub> pollution, whereas the natural sources, like dust storms, are estimated to cause only about 22% of such mortality (Yang et al., 2022) and are more difficult to control. Purposeful short-term policy interventions (e.g., the 2008 Beijing Olympics, the 2014 Asia-Pacific Economic Cooperation Summit) have yielded improvement of air quality at local scales and during the targeted events (W. Wang et al., 2009; X. Wang et al., 2009; Chen et al., 2013; Liu and Ogunc, 2023). However, it is also revealed that such local-scale emission controls could be compensated by increases of emissions in surrounding areas, highlighting the importance of regional emission controls in achieving improved air quality in a city (Wang et al., 2010). Previous studies that carried out global model experiments have shown potentially significant albeit highly uncertain effects of a hypothetical and uniform 20% reduction of global anthropogenic emissions on air quality, climate, and ecosystems at local, regional, and intercontinental scales (e.g., Shindell et al., 2008; Yu et al., 2013; Collins et al., 2013; Anenberg et al., 2014).

The COVID-19 lockdowns created a natural laboratory for studying the effect of reducing anthropogenic activities on urban air quality at a global scale and during an extended period. Since the emergence of the novel coronavirus disease 2019 (COVID-19) in early 2020, governments around the globe had enforced a variety of measures, including regional and national lockdowns, to contain rapid spread of the virus and protect the wellness of human beings. There is no doubt that these lockdown measures reduced emissions of various anthropogenic pollutants and greenhouse gases. The science community promptly seized this opportunity to investigate the changes of air pollution following the lockdowns. Over a short period of less than two years since the lockdowns, a large body of studies had already been published (see reviews in Gkatzelis et al., 2021; Laughner et al., 2021; Saha et al., 2022; Bakola et al., 2022). The examined atmospheric pollutants included nitrogen dioxide (NO<sub>2</sub>), PM<sub>2.5</sub>, and ozone (O<sub>3</sub>), among others. In general, effects of the COVID-19 lockdown have been derived in the previous studies by comparing concentrations of air pollutants during the lockdown period against those immediately prior to the lockdown (e.g., Shi and Brasseur, 2020) or against the climatology during the same period in pre-pandemic years (e.g., Venter et al., 2020). Modeling studies driven by the reduced emissions caused by the COVID-19 lockdowns have also been performed (Miyazaki et al., 2020; Le et al., 2020). These studies have consistently showed reductions in NO2, a short-lived pollutant with a major source from the transportation sector that is considered as a good proxy for emissions, during the lockdowns (Liu et al., 2020). However, the change of surface ozone concentrations during the lockdowns was reported to increase in many locations despite the widespread reduction of its NO<sub>2</sub> precursor (e.g., Shi and Brasseur, 2020; Venter et al., 2020; Shi et al., 2021; Le et al., 2020), although one study found a global scale decline in ozone burden using satellite observations (Miyazaki et al., 2021). For PM<sub>2.5</sub>, the effects of lockdown have showed mixed results (Venter et al. 2020; Shi et al., 2021; Volta et al., 2022; Putaud et al., 2023), suggesting the challenge in discerning and quantifying the effects induced by the reductions of anthropogenic emissions associated with lockdowns. Accounting for the effects of variability of meteorological conditions, anthropogenic emission trends, and contributions of natural sources of PM<sub>2.5</sub> that is not affected by

the lockdown is crucial to addressing the challenge and reliably assessing the effect of lockdowns (Gkatzelis et al., 2021; Shi et al., 2021; Le et al., 2020). Several empirical approaches of de-weathering and de-trending have been used in some previous studies. For example, Volta et al. (2022) attempted to account for meteorological impacts on air pollution by classifying favorable and unfavorable meteorological conditions for air quality and comparing them between the pandemic and prepandemic years. The linear regression of pre-pandemic air quality data has been applied to account for the air pollution trend due to clean air policies (Volta et al., 2022). Multivariate regression analysis and machine learning approaches have also been used to predict air quality in the pandemic year in a no-lockdown scenario (Venter et al., 2020; Shi et al., 2021; Anderson et al., 2021; Ghahremanloo et al., 2022).

The objective of this study is to improve the understanding of COVID-19 lockdowns' effects on PM<sub>2.5</sub> air quality through a synergistic analysis of observational data and model simulations. Our study focuses on surface-level PM<sub>2.5</sub> because it has the most detrimental effect to human health among all the pollutants (Cohen et al., 2017). We hypothesize that the overall reductions in anthropogenic emissions from many sectors brought about by the COVID-19 lockdowns would have improved the PM<sub>2.5</sub> air quality on a global scale. However, such impacts might have been masked by other factors such as meteorological conditions and natural emissions (e.g., dust storms, wildfires) in the observational datasets. The impacts might also be determined by the relative contributions of individual sectors because some sectors might have increased the emissions during lockdowns. To test the hypothesis, we analyze multi-year surface PM<sub>2.5</sub> observations in urban areas around the globe to examine the potential anomaly of PM<sub>2.5</sub> concentrations during the major lockdown periods with respect to that in the prepandemic years. Then we use modeling experiments to disentangle the effect of the lockdown emission reductions from non-lockdown effects. We also investigate if satellite remote sensing observations of aerosol optical depth (AOD) can be used to identify the change of surface PM<sub>2.5</sub> due to the COVID-19 lockdowns.

The rest of paper is organized as follows. In section 2, we provide a brief description of observational and the Goddard Earth Observing System (GEOS) modeled datasets (such as PM<sub>2.5</sub>, AOD) and an auxiliary stringency index for identifying the dynamics of the lockdowns around the world. Results of data analysis are presented in Section 3, including the evidence of coincidence of declining PM<sub>2.5</sub> with increasing stringency index, comparisons of observed and modeled changes in PM<sub>2.5</sub> in 2020 in the context of climatology, as well as GEOS-based relative contributions of the lockdown emission reductions and non-lockdown factors in explaining the difference between 2020 and 2019. In Section 4, we discuss some remaining issues associated with the analysis and an investigation of the feasibility of identifying the lockdown effects on PM<sub>2.5</sub> from the satellite AOD measurements. Major conclusions are summarized in Section 5.

#### 2. Descriptions of observational data and model simulations

## 2.1. Observations of surface PM<sub>2.5</sub> and aerosol optical depth

For the surface PM<sub>2.5</sub>, we use observations collected from the AirNow Department of State network, which include sites at the USA diplomatic posts (i.e., Embassies and Consulates) in major cities outside the USA

(https://www.airnow.gov/international/us-embassies-and-consulates/). We selected 17 posts around the world with observations over a period of at least five years, which includes the pandemic years (2020-2022) and at least two pre-pandemic years. The number of pre-pandemic years depends on the diplomatic post, as the State Department has been gradually extending the PM<sub>2.5</sub> monitoring from the diplomatic posts in China to those in India and then other countries. Because most of the USA diplomatic posts equipped with multi-year PM2.5 measurements are in East Asia, South Asia, Middle East, South America, and North Africa, we also included PM<sub>2.5</sub> observations in four additional cities from other national air quality networks to extend the representativeness of our analysis to North America and Europe. Paris, France and Milano, Italy were selected to represent Europe, while New York City, New York and Los Angeles, California were chosen to represent North America. Figure 1 illustrates the geographical distribution of 21 urban stations in 13 countries with PM<sub>2.5</sub> observations, with detailed information (including city and country names, longitude, latitude, and years of data used in the analysis) listed in the supplement (*Table S1*).

These 21 stations are representative of distinct aerosol characteristics, as shown in Figure 2 for the fractional contributions to annual mean PM<sub>2.5</sub> by an array of source sectors. The source-sector partitions were based on the GEOS-Chem sector sensitivity 115 simulations for year 2017 as provided in McDuffie et al. (2021). This modeling used global anthropogenic emission inventory for seven key pollutants (nitrogen oxides - NO<sub>x</sub>, sulfur dioxide - SO<sub>2</sub>, carbon monoxide - CO, ammonia - NH<sub>3</sub>, non-methan volatile organic carbons - NMVOCs, black carbon - BC, and organic carbon - OC) from 11 anthropogenic sources and four fuel types, which was developed from the Community Emissions Data System (CEDS) with updates for the Global Burden of Disease Major Air Pollution Sources project (CEDS GBD-MAPS) (McDuffie et al.. 2020, https://zenodo.org/records/3754964). Additional emission inputs to GEOS-Chem include those from fires, biogenic sources. and anthropogenic and desert dust, as described in the supplementary Table 2 of McDuffie et al. (2021) and references therein. Figure 2 clearly displays that sector contributions to PM<sub>2.5</sub> vary substantially from station to station. Among the source sectors considered in the GEOS-Chem modeling, emissions from energy, industry, transportation, commercial and other combustions, international shipping, and anthropogenic fugitive, combustion, and industrial dust (AFCID) might have decreased during the lockdowns due to reduced human mobility, with magnitudes of decrease likely depending on the specific sector. For brevity, we refer to these six sectors collectively as to potential lockdown emission reduction sectors (LERS). Presumably, the

transportation sector had the largest reduction in emissions during lockdown. On the other hand, emissions from the residential sector would have increased during the pandemic because of the "work/study from home" during the lockdown period that should lead to more extended usage of electricity, heating and cooling at home. Natural events like desert dust storms and wildfires were likely unaffected by the lockdowns, although anthropogenic emission reductions associated with the lockdowns may have impacted dust and fire weather to some unquantified extent. An occurrence of such large episodic natural events could even mask the effect of anthropogenic emission reductions associated with the lockdowns on PM<sub>2.5</sub> air quality. The modeling-based PM<sub>2.5</sub> source characteristics, albeit inevitably subject to large uncertainties (McDuffie et al., 2021), could facilitate a qualitative interpretation of observed changes in PM<sub>2.5</sub> in response to the lockdowns discussed in Section 3. In general, when the fractional contributions by the LERS are high and those by residential sectors are low, the lockdown's impacts could be more clearly shown in observed PM<sub>2.5</sub> data. However, when PM<sub>2.5</sub> in a city is dominantly sourced from episodic events such as desert dust and wildfires, the lockdown's signals might be masked by these events.

1 – Beijing; 2 – Shenyang; 3 – Shanghai; 4 – Guangzhou; 5 – Hanoi; 6 – Jakarta; 7 – Chennai; 8 – New Delhi; 9 – Mumbai; 10 – Kolkata; 11 – Hyderabad; 12 – Addis Ababa; 13 – Dubai; 14 – Kuwait City; 15 – Manama; 16- Paris; 17 – Milan; 18 – Pristina; 19- New York City; 20 – Los Angeles; 21 - Lima

Figure 1: Geographical distributions of PM<sub>2.5</sub> observational stations in 21 cities (open and solid circles, which are numbered from 1 to 21 with names listed below the figure) of 13 countries overlying on the GEOS simulated PM<sub>2.5</sub> concentrations (colored contours, with a unit of µg m<sup>-3</sup>) in March 2020. Solid black circles, corresponding to site numbers of 3, 8, 14, 16, 20, and 21, denote the six cities being selected for in-depth analysis and representative of distinct aerosol characteristics. Detailed information about all these cities is listed in Table S1 of the supplement, including city and country names, latitude, longitude, and the period of PM<sub>2.5</sub> data used in this study.

145

Figure 2: Fractional contributions to PM<sub>2.5</sub> in 21 stations by source sectors, with a total percentage (%) of the six lockdown emission reduction sectors (LERS, including energy, industry, transportation, commercial & other combustion, AFCID dust, and shipping) denoted in the parenthesis immediately after the station name. The chart was made with the sector contributions derived from the GEOS-Chem sensitivity simulations for 2017 and provided by the supplementary material of McDuffie et al. (2021) and https://zenodo.org/records/4739100.

Even with the same emissions, the surface PM<sub>2.5</sub> concentrations can vary greatly from day to day because of strong regulation by rapidly evolving meteorological and chemical processes. It is oftentimes formidable to discern any meaningful changes in PM<sub>2.5</sub> during the pandemic years relative to the pre-pandemic years on a daily basis. In this study, we calculate 5-day running means of PM<sub>2.5</sub> to smooth out high-frequency variations so that potential differences between the pandemic years (2020, 2021, and 2022) and pre-pandemic years (prior to 2020) could show up in the observations more clearly. A sensitivity test of using 3-day or 7-day running means suggests that the major conclusions of this study will not change. To obtain quantitative estimates of PM<sub>2.5</sub> changes due to prolonged lockdowns, we also compute the monthly average PM<sub>2.5</sub> for both the pandemic years and pre-pandemic years.

AOD observations from the Moderate Resolution Imaging Spectroradiometer (MODIS) on the Aqua satellite (Levy et al., 2013) were also used in this study. Unlike surface PM<sub>2.5</sub> that measures the concentration of air pollution at our nose level, AOD measures the load of aerosol in the whole column of the atmosphere. Because of the appealing nature of routine satellite observations at a global scale, numerous studies have explored the use of satellite AOD to derive surface PM<sub>2.5</sub> concentrations (e.g., van Donkelaar et al., 2010; Wei et al., 2021). However, the relationship between AOD and PM<sub>2.5</sub> is complicated by several factors, such as aerosol composition, vertical profile of aerosol, relative humidity of ambient atmosphere, and atmospheric long-range transport. In this study, we examine if satellite AOD measurements can be used to detect the change in PM<sub>2.5</sub> due to the COVID lockdowns.

## 2.2. GEOS simulations of PM<sub>2.5</sub> and AOD

Simulations from the NASA GEOS model are used for both AOD and PM<sub>2.5</sub> in this study. The modular GEOS model is a global Earth system model that includes components for atmospheric circulation and composition, ocean circulation and biogeochemistry, and land surface processes (Rienecker et al., 2011; Molod et al., 2015). The coupled atmospheric constituent module within the GEOS architecture most relevant to this project is an aerosol module based on the Goddard Chemistry Aerosol Radiation Transport (GOCART) model (Collow et al., 2024). GOCART simulates major components of aerosols (with a diameter between 0.02 and 20 µm) and several gaseous precursors, including dust, sea-salt, sulfate, nitrate, organic carbon, black carbon, SO<sub>2</sub>, and dimethyl sulfide (Chin et al., 2002, 2007, 2009, 2014; Ginoux et al., 2001; Bian et al., 2017). The model considers the atmospheric processes of chemistry, convection, advection, boundary layer mixing, dry and wet

deposition, and gravitational settling (Chin et al., 2002, 2014). Aerosol particle sizes are simulated with parameterized hygroscopic growth, which is a function of ambient relative humidity. Total mass of sulfate and carbonaceous aerosols are calculated, while nitrate aerosol mass is calculated in three bins (Bian et al., 2017). For dust and sea salt, the particle size distribution is explicitly resolved across five size bins (Chin et al., 2002, 2009).

For this study, the GEOS model is run at a horizontal resolution of 0.5° for 2019 (pre-pandemic year) and 2020 (the first year of the pandemic). The required meteorological fields are taken from the Modern-Era Retrospective analysis for Research and Applications - version 2 (MERRA2, Gelaro et al., 2017). For 2019, anthropogenic emissions were taken from an updated version (V\_2021\_04\_21) of the Community Emission Data System (CEDS) (Hoesly et al., 2018). For 2020, we carry out two modeling experiments, denoted as 2020-BAU and 2020-COVID, by using two anthropogenic aerosol and precursor emissions. In the 2020-BAU scenario, we used the anthropogenic emissions for 2019 to approximate the business-as-usual (BAU) anthropogenic emissions for 2020. In the 2020-COVID scenario, the 2019 anthropogenic emissions from different sectors were adjusted based on daily mobility data gathered by Apple and Google to reflect the lockdown's effects on the anthropogenic emissions (Foster et al., 2020). Note that in both 2020-BAU and 2020-COVID runs, emissions from desert dust, sea-sprays, and wildfires are representative for 2020 conditions; specifically, emissions from wildfires are prescribed based on satellite observations in 2020 (Darmenov and da Silva, 2015) and dust and sea-salt emissions are calculated online within the GEOS model based on surface and meteorological fields in 2020 from the MERRA-2 analysis (Chin et al., 2014).

One of the advantages of using GEOS modeling is that it provides not only total AOD and PM<sub>2.5</sub> but also their composition. This allows for distinguishing anthropogenic sources from natural sources (e.g., dust storms, wildfires, volcanic eruption or degassing, and sea sprays). In addition to providing the 10-year pre-pandemic climatology of PM<sub>2.5</sub> and AOD, the two experiments for year 2020 (i.e., 2020-BAU and 2020-COVID) together with the 2019 run can be used to distinguish the effects of anthropogenic emission reductions associated with the COVID-19 lockdowns from those associated with differing meteorological conditions (through affecting aerosol transport and removal processes) as well as effects from emissions of natural aerosols, which are not directly related to the lockdowns. For brevity, we refer to the differences between 2020-COVID and 2020-BAU as the "lockdown effect" whereas the difference between 2020-BAU and 2019 is referred to as the "non-lockdown effect" that is due to changes in meteorological conditions and natural emissions. These experiments can be used to

facilitate the interpretation of the observed PM<sub>2.5</sub> difference between 2020 and 2019. In this study, it is assumed that the differing meteorology and natural emissions between 2020 and 2019 are not caused by the lockdown-induced anthropogenic emissions.

## 2.3. Stringency index measuring the scope of lockdowns

The rapid spread of COVID-19 promoted a wide range of government responses in containing the disease and protecting the
wellness of human beings. The Oxford COVID-19 Government Response Tracker (OxCGRT) project was established to track
the policy indicators of government responses at national and even state/province levels. The project remained active for most
of the nations tracked until the end of 2022. The OxCGRT provided four composite indices by grouping different families of
policy indicators, namely the government response index, the stringency index, the containment and health index, and the
economic support index (Hale et al., 2021). A stringency index, measuring the severity of lockdown restrictions, was developed
by aggregating nine policy indicators, including school closures, workplace closures, public event cancellations, restrictions
in gathering size, and travel bans, among others. The index is rescaled to a number between 0 and 100, with 100 being the
most extreme lockdown situation (Hale et al., 2021). In this study, we use the stringency index to identify major lockdown
periods to facilitate the analysis of change in PM<sub>2.5</sub> air quality. Although for 11 stations in China, India, and USA the stringency
indices were derived from lockdown measures implemented at the state/province level, the stringency indices for the other 10
stations were determined based on national lockdown measures.

#### 3. Results

In this section, we first present a detailed analysis of PM<sub>2.5</sub> change in response to the lockdowns on the 5-day running mean and monthly mean basis in six populated major cities (marked as filled black dots in Figure 1), namely Shanghai (China), New Delhi (India), Los Angeles (USA), Paris (France), Lima (Peru), and Kuwait City (Kuwait). These stations are selected to represent broad geographical regions with different aerosol characteristics in terms of source sectors (as shown in Figure 2), which would determine how PM<sub>2.5</sub> levels responded to the COVID-19 lockdowns. Then we interpret the observed changes in PM<sub>2.5</sub> in the context of regional PM<sub>2.5</sub> trends and attribute them to the COVID-lockdown induced anthropogenic emissions reductions and the interannual variability of meteorological conditions with the aid of GEOS modeling experiments. Finally,

we present a general discussion of observed PM<sub>2.5</sub> changes during the major lockdown periods in the remaining 15 stations (open circles in Figure 1).

# 3.1. Observed changes in PM<sub>2.5</sub> corresponding to COVID lockdowns – Case studies of selected stations

Figures 3-8 display PM<sub>2.5</sub> variations at the six stations on the 5-day running mean and monthly mean basis. For each monitoring station, shown in the top panel (a) is the time series of stringency index during 2020 – 2022 period, which can be used to identify the lockdown periods and facilitate the interpretation of observed PM<sub>2.5</sub> variations, whereas shown in panel (b) and (c) are the observed time series of 5-day running mean and monthly average of PM<sub>2.5</sub>, respectively, in pre-pandemic years prior to 2020 (the number of years depending on available observations at the station) and during the pandemic years of 2020-2022. The pre-pandemic (2019 and earlier) climatologies are represented by the average (black lines) and the range of observations (gray vertical bars). We also display the time series in 2019, a year immediately prior to the pandemic, laying the ground for a focused analysis of 2020 and 2019 in the next section. Major characteristics of the PM<sub>2.5</sub> level in response to the COVID-19 lockdowns are detailed in the following subsections. We particularly focus on PM<sub>2.5</sub> changes during relatively stringent lockdown periods, defined in this study as periods with a stringency index > 60.

## 3.1.1. Shanghai, China

Shanghai's lockdowns began in late January 2020, when its stringency index increased from almost 0 to more than 70 on January 27 (day of the year or DOY = 27), as shown by the red line in Figure 3a. Moderately strict lockdown regulations continued through the rest of this year, seeing that values for the stringency index remained above 45. The station's observed 5-day moving average concentrations of PM<sub>2.5</sub> in 2020 were generally lower than that in 2019 and other pre-pandemic years. There was a sudden drop of the 2020 (red line) average from the pre-pandemic multi-year average (black line) in Figure 3b when DOY = 27, and a clear disparity between the two lines continues throughout the year except a period of DOY=196-265 (mid-July to late-September). Similarly, monthly PM<sub>2.5</sub> levels portrayed in Figure 3c from February through December 2020 were significantly lower than the pre-pandemic averages except July-September when differences became smaller. The most evident pullback from years preceding the pandemic to 2020 seems to be in March, which was a relative reduction of 55% in comparison to the pre-pandemic average. This matches with how Shanghai's stringency index was at its largest for that year

from February through May. Thus, it is reasonable to assume that Shanghai's lockdowns during this time contributed to lowering the PM<sub>2.5</sub> levels.

Furthermore, Shanghai's second major lockdown began around April 2022 that was even stricter than the 2020 lockdown with the stringency index >90. The blue line in Figure 3a reaches a peak of 97, an almost maximum stringency, and only falls to approximately 60 at the start of June. During these months of regulations triggered by the Chinese government's zero-COVID policy, where rapid lockdowns and mass testing would occur whenever positive cases emerged, and Shanghai's 26 million citizens were mostly confined in their homes (Han et al., 2024). As indicated in Figure 3c, Shanghai's 2022 monthly PM<sub>2.5</sub> means during April, May, and June 2022 reached record lows, staying well below the means of all past years. In particular, the April 2022 average was less than the 2012-2019 average by about 35 µg m<sup>-3</sup>. On the monthly basis, PM<sub>2.5</sub> from March to May decreased from pre-pandemic means by 43% to 56%. In Figure 3b, 5-day moving averages for 2022 also occasionally dipped to historical minimums once in late April and twice in May. It is evident that Shanghai's unyielding shutdown of 2022 that practically locked its population indoors had a pronounced impact on PM<sub>2.5</sub> levels. In summary, the above data analysis shows that the COVID-19 lockdowns reduced the PM<sub>2.5</sub> load, consistent with the fact that 60% of PM<sub>2.5</sub> in Shanghai came from those potential LERS (Figure 2) as discussed in section 2.1.

#### 3.1.2. New Delhi, India

In New Delhi, it is estimated that about 47% of PM<sub>2.5</sub> were sourced from those potential LERS, while residential sector accounted for about 27% (Figure 2). Based on the time series in **Figure 4a**, New Delhi's government started imposing stricter shutdown measures in February of 2020, and went into a full lockdown by mid-March that lasted until the end of May, where the stringency index hit 100. From June through September 2020, the stringency index gradually decreased yet was always above 60, meaning that New Delhi continued moderate COVID-19 regulations for most of the year. Comparing these values with the 5-day running means of PM<sub>2.5</sub> in Figure 4b, it is likely that the COVID-19 lockdowns had at least some impact on air quality in New Delhi. The red line of PM<sub>2.5</sub> in 2020 falls under the black line of pre-pandemic (2015-2019) averages from around DOY = 90 to DOY = 135, demonstrating lowered PM<sub>2.5</sub> in April and part of May, when lockdown measures were at maximum severeness. The monthly PM<sub>2.5</sub> levels for March through May 2020 in Figure 4c were also lower than the

corresponding 2015-2019 monthly means by 29% to 45%. In the later months of 2020, when shutdown restrictions lessened, monthly average PM<sub>2.5</sub> concentrations became similar to those in past years.

In the spring of 2021, New Delhi experienced a second, smaller-scale COVID-19 lockdown. According to the stringency index graph, the yellow line representing the city's index in 2021 up ticked to values between 80 and 100 from DOY = 109 to DOY = 158. New Delhi's PM<sub>2.5</sub> levels, however, did not have any noticeable decrease or change in the early stages of this timeframe, as the March and April 2021 means were almost the same magnitude as or even higher than the same means of pre-pandemic years. But by the end of the lockdown period, the monthly PM<sub>2.5</sub> concentration dropped to a record low (October 2021). The May 2021 average PM<sub>2.5</sub> dropped to as small as the May 2020 mean. Despite this occurrence, there is no strong indication that the 2021 lockdown in New Delhi had much influence on air pollution, because the amount of PM<sub>2.5</sub> in its atmosphere only decreased during less than half of the highly stringent period.

## 3.1.3. Los Angeles, USA

For Los Angeles, 58% of PM<sub>2.5</sub> in pre-pandemic years were sourced from potential LERS. The reductions in PM<sub>2.5</sub> from these
LERS during the pandemic years might be compensated by an increase associated with the residential sector, which accounted
for about 17% of pre-pandemic PM<sub>2.5</sub>. COVID-19 lockdown measures were strictest when the stringency index quickly
elevated to above 80 in mid-March 2020, specifically on DOY = 79, from an initial value of around 20 ten days earlier (Figure
5a). The index plateaued in the low 80s through April of 2020. In this timespan, PM<sub>2.5</sub> concentrations diminished to some
extent, with the red line of 2020 means remaining under the black line of pre-pandemic means throughout March and the
beginning of April (Figure 5b). Moreover, we show in Figure 5c that March 2020 and April 2020 PM<sub>2.5</sub> averages were smaller
than their pre-pandemic counterparts by 47% and 32%. Several steep spikes in 5-day moving averages of PM<sub>2.5</sub> levels also
occurred in the 2020 summer and autumn (Figure 5b), due to the extreme wildfires that struck California that year, which
burned about 4.3 million acres of land total, a number twice the state's previous record (Safford et al., 2022).

## 3.1.4. Paris, France

Paris experienced lockdowns induced by COVID-19 in 2020 and 2021. In 2020, the city's stringency index reached 88 at DOY = 77 (mid-March) and stayed at a similar level until around DOY = 131 (mid-May) in Figure 6a. Interestingly, the PM<sub>2.5</sub> level hit a record low in many days in February, prior to the lockdown (Figure 6b). For the rest of the year, the 5-day moving

average of the PM2.5 level stayed below the 2013-2019 mean for most days, although it became closer to the 2019 level starting from July after the lockdown measures were eased. For the monthly averages in Figure 6c, PM2.5 values in March and April 2020 decreased from the 2013-2019 means by approximately 43% and 13%, respectively, although it is noticed that the PM2.5 in February (before lockdown) was the lowest in the entire year. Unlike other sites discussed earlier in this paper, the PM2.5 levels in Paris rarely hit record lows during lockdowns, even though they are near the bottom of the range from prepandemic years.

Another less rigid lockdown for Paris went into place during January to mid-May of 2021, where the stringency index lay between 60 and 80. This time, the PM<sub>2.5</sub> concentrations were also lower than the pre-pandemic values (except April) but higher than in 2020. This is more clearly seen in the monthly means, as the PM<sub>2.5</sub> in 2021 from January to May are lower than the same months before the pandemic but are higher than 2020 except May, apparently consistent with the different degrees of the stringency between 2020 and 2021, although such attribution might be ambiguous.

#### **3.1.5.** Lima, Peru

305

310

Lima's most strict lockdown occurred mid-March through June of 2020, as seen in Figure 7a. In these nearly four months, the stringency index ranged from 90 to 97, a sign of extremely tight lockdown measures. Evidently, in Figure 7b, the PM<sub>2.5</sub> levels for 2020 fell well below 2016-2019 amounts during the exact same period. Monthly averages of PM<sub>2.5</sub> in Lima rendered by Figure 7c were also significantly diminished in March to June of 2020, where concentrations consistently were lower than prepandemic averages by 47% to 54%. Since the timings of declines in PM<sub>2.5</sub> matched with the timings of escalated stringency indices, it seems to present a line of strong evidence that the reduction of human activities had direct impact on lowering the PM<sub>2.5</sub> levels in Lima. Such effects are more clearly seen in the monthly mean PM<sub>2.5</sub> in Figure 7c that the 2020 values are lower than the pre-pandemic years throughout the year with the largest reduction in the months with highest stringency index.

In contrast, the lockdown measures in 2021 did not seem to help reducing the PM<sub>2.5</sub> in Lima despite the still high stringency index (around 60) throughout the year. The PM<sub>2.5</sub> levels in 2021 were very similar to the pre-pandemic values, although they were noticeably lower by 6.5 and 10.3 μg m<sup>-3</sup> (corresponding to 30% and 41%) in February and March of 2021 respectively (Figure 7b and 7c) when the stringency index was higher (~70-75).

## 3.1.6. Kuwait City, Kuwait

During April, May, June, and July of 2020, Kuwait City's stringency index enlarged to values between 80 and nearly 100 (Figure 8a). Despite this severe lockdown, there were not many apparent signs of change in PM<sub>2.5</sub> concentrations in the spring and summer of 2020, which often surpassed pre-pandemic concentrations and occasionally fell to record lows. Figure 8b's red line for PM<sub>2.5</sub> 5-day moving averages in 2020 does reach minimums under the black line of pre-pandemic averages at points in mid-April and early May, and monthly means in Figure 8c for March and April 2020 were less than their complementary 2017-2019 means by 13% to 40%. Nonetheless, the concentration of PM<sub>2.5</sub> in 2020 in general were well within the range of pre-pandemic years. In 2021, although the stringency index remained above 60 from January to July, the PM<sub>2.5</sub> levels did not respond to the stringency measures (Figure 8b and 8c). These levels hit the record low in August-October despite the relaxed stringency.

It should be noted that Kuwait City is heavily influenced by desert dust, which was not affected by the lockdown. Therefore, it is expected that dust frequently plays a determining role in PM<sub>2.5</sub> levels in Kuwait City. This is most evident in 2022 (blue lines in Figures 8b and 8c) when the PM<sub>2.5</sub> concentrations were pulsed and abnormally large with the May average being close to twice as high as the mean for May from pre-pandemic years. The heavy dust storms that occurred in late May across the Middle East in that year (Francis et al., 2023) are a probable explanation for this growth. It is logical to assume then, that this event is proof of how the presence of dust storms can have a magnified effect on the overall levels of PM<sub>2.5</sub> in Kuwait City, which would overwhelm changes due to other influences like COVID-19 lockdowns, in observational data.

Figure 3: Time series of (a) the lockdown stringency index during the pandemic years (2020 - red, 2021 - orange, and 2022 - blue) and (b) the 5-day running mean PM<sub>2.5</sub> concentration over years for four individual years of 2019-2022 (with brown, red, orange, and blue line denotes 2019, 2020, 2021, and 2022, respectively) in Shanghai, China. The monthly average PM<sub>2.5</sub> concentrations (μg m<sup>-3</sup>) are shown in (c). Also shown in (b) and (c) is the pre-pandemic climatology with black line denoting the average and gray shaded areas the range of observations during the pre-pandemic period.

Figure 4: Same as Figure 3, except for New Delhi, India.

Figure 5: Same as Figure 3 except for Los Angeles, USA.

Figure 6: Same as Figure 3, except for Paris, France.

Figure 7: Same as Figure 3, except for Lima, Peru.

Figure 8: Same as Figure 3, except for Kuwait City, Kuwait.

## 3.2. Attribution of the observed changes in PM<sub>2.5</sub> between the pandemic and pre-pandemic years

In this section, we attempt to interpret the observed changes in PM<sub>2.5</sub> by carrying out additional analyses of in-situ PM<sub>2.5</sub> observations and GEOS model simulations to address two major questions: (1) are the observed changes in PM<sub>2.5</sub> during the lockdowns coincident with the regional PM<sub>2.5</sub> trends? (2) what are the relative contributions of the COVID-lockdown induced anthropogenic emission reductions, interannual variability of meteorology, and natural emissions to the PM<sub>2.5</sub> changes during the pandemic?

## 3.2.1. Inferring the Lockdowns' impacts in the context of regional PM<sub>2.5</sub> trends

375

The analysis in section 3.1 showed that on a monthly basis, PM<sub>2.5</sub> in 2020 during the lockdown periods was systematically lower than the pre-pandemic averages in six cities except at locations significantly affected by wildfires and dust storms. In some cities, the 2020 PM<sub>2.5</sub> concentrations even made it to record lows in the study periods. However, these reductions in PM<sub>2.5</sub> during the pandemic may not be simply credited to the effects of COVID-19 lockdowns. For instance, if the PM<sub>2.5</sub> level in a city had been declining before the pandemic, an apparent decrease in PM<sub>2.5</sub> observed in 2020 might be just following the trend in a business-as-usual scenario. It is thus necessary to put them in the context of regional PM<sub>2.5</sub> interannual variations or trends to reduce ambiguity in attributing the observed PM<sub>2.5</sub> reductions to the lockdown impacts.

Here we investigate the March-April average PM<sub>2.5</sub> concentrations in pre-pandemic years (black dots) and during the pandemic years (red triangles) for all six cities, as shown in **Figure 9**. We restricted the period to March and April because this was when the first wave of lockdowns occurred at most of the cities in 2020. These plots also contain standard deviations and linear regression lines (black lines) of the data for all years up until 2019 (i.e., excluding the pandemic years 2020-2022). The coefficients of determination, R<sup>2</sup>, of the linear regressions for Shanghai and Paris is 0.739 and 0.785, respectively, which suggests that the pre-pandemic decreasing trend is statistically significant with p = 0.01 based on the student t-test. Although R<sup>2</sup> for Lima and Kuwait City are higher, the limited data points in the PM<sub>2.5</sub> observations make the R<sup>2</sup> values less meaningful in terms of statistical significance of the decreasing trend. Contrary to the other four locations, PM<sub>2.5</sub> levels in New Delhi and Los Angeles contained no meaningful trends, as R<sup>2</sup> values for both sets of data points are 0.1 or less. The linear regression lines based on the pre-pandemic PM<sub>2.5</sub> observations are then extrapolated to later years as an estimation of PM<sub>2.5</sub> under a BAU scenario in 2020-2022 without COVID-19 lockdown. A deviation of observed PM<sub>2.5</sub> levels during the pandemic years from

the predicted values is compared with the pre-pandemic standard deviation to assess the likelihood of influences by the COVID-19 lockdown. For Shanghai, the deviation of observed PM<sub>2.5</sub> from the predicted value is -13.73, -7.07, and -9.94 μg m<sup>-3</sup> for 2020, 2021, and 2022, respectively. For comparison, the pre-pandemic standard deviation is 7.33 μg m<sup>-3</sup>. This suggests that emission decreased associated with Shanghai's stringent pandemic restrictions most likely have caused significant reductions of PM<sub>2.5</sub> in 2020 and 2022. Similarly, the deviation from the predicted value in 2020 is -30.94, -5.44, and -6.38 μg m<sup>-3</sup> in New Delhi, Los Angeles, and Lima, respectively. These deviations are significantly greater than the corresponding pre-pandemic standard deviations of 6.81, 2.58, and 3.89 μg m<sup>-3</sup>, most likely suggesting that the COVID-19 lockdowns in 2020 caused significant reduction of PM<sub>2.5</sub> in New Delhi, Los Angeles, and Lima. On the other hand, 2020's PM<sub>2.5</sub> levels in Paris and Kuwait City were above the regression line by 2.55 and 4.12 μg m<sup>-3</sup>, respectively, despite being lower than prepandemic averages. In these cases, external causes may have offset any impacts of reduced anthropogenic emissions during COVID-19 lockdowns.

Notably, in March-April of 2021 and 2022, all six cities except for Shanghai demonstrated upticks in PM<sub>2.5</sub> concentration to a level near or well above the BAU projection. Shanghai's PM<sub>2.5</sub> increased in 2021 but decreased again in 2022 due to the second stringent lockdown discussed earlier. The PM<sub>2.5</sub> concentration in 2022 was even lower than that in 2020. The question of how much the loosening of COVID-19 pandemic regulations truly contributed to these increases in PM<sub>2.5</sub> requires further investigation, but so far, there is some evidence of elements besides the lockdown stringency, such as the dust storms in Kuwait City, playing a role as well.

Figure 9: Interannual variations of the March-April average PM<sub>2.5</sub> (μg m<sup>-3</sup>) in (a) Shanghai, (b) New Delhi, (c) Los Angeles, (d) Paris, (e) Lima, and (f) Kuwait City. Black dots indicate pre-pandemic years and red dots for the pandemic years. Black line represents a linear regression of PM<sub>2.5</sub> for pre-pandemic years only (excluding data points in the pandemic years), which is used to predict the business-as-usual (BAU) PM<sub>2.5</sub> during the pandemic years. R<sup>2</sup> and pre-pandemic standard deviation (σ) is noted in the box. Red triangles indicate average PM<sub>2.5</sub> in the pandemic years.

3.2.2. Attributing the observed PM<sub>2.5</sub> changes to the lockdown-induced emission reductions with GEOS modeling 425 In this section, we focus on an analysis of the observed and modeled changes in March-April average PM<sub>2.5</sub> concentration between 2020 and 2019. This focused analysis is done based on the three GEOS model experimental runs (as described earlier) to provide insight into how the observed PM<sub>2.5</sub> changes are related to the anthropogenic emission reductions associated with the COVID-lockdowns and differing meteorology/natural emissions. The three GEOS runs for 2019, 2020-BAU, and 2020-430 COVID are compared against each other to distinguish a change associated the lockdowns (i.e., reductions in anthropogenic emissions) from those non-lockdown effects (e.g., interannual variations in meteorological conditions and natural emissions). Specifically, the difference between 2020-BAU and 2019 indicates the effect of differing meteorology and natural emissions between 2020 and 2019, as the same anthropogenic emissions were applied in these two runs. On the other hand, the difference between 2020-COVID and 2020-BAU measures the effect of reducing anthropogenic emissions by the lockdown restrictions, as the two runs were driven by the same meteorology and emissions of natural aerosols from dust storms, sea sprays, 435 biogenic/volcanic sources, and wildfires were largely the same. These simple attributions are insightful, although the soderived lockdown effects and those by the differing meteorology and natural emissions (i.e., non-lockdown effects) may not

add up exactly to the reduction in PM<sub>2.5</sub> due to the nonlinearity of the aerosol system.

**Table 1** lists the relative changes (%) of anthropogenic SO<sub>2</sub>, NH<sub>3</sub>, BC, and OC emissions in the six cities due to implementation of the lockdowns, on a basis of March-April average (annual cycles of BAU and COVID emissions are shown in the supplement, *Figures S3-S8*). We derived these numbers by calculating differences of March-April emissions around the six cities (averaged over a 3° x3° box around each city) between 2020-COVID and 2020-BAU scenarios of anthropogenic emissions that were used to drive the GEOS simulations. As described in section 2.2, for the 2020-BAU scenario, anthropogenic emissions in 2019 were used to represent the baseline emissions of 2020, assuming the anthropogenic emissions would not have significant changes from 2019 to 2020 in a business-as-usual (BAU) scenario. For the 2020-COVID scenario, the 2019 anthropogenic emissions in individual sectors were adjusted (decreased or increased, depending on sectors) based on daily mobility data gathered by Apple and Google to reflect the COVID lockdown's impacts on anthropogenic emissions, which was developed by Foster et al. (2020). While the anthropogenic emissions generally decreased because of the lockdowns by a large range of magnitudes (-2.1% to -45.9%) depending on locations and species, OC in New Delhi increased slightly by

+1.5%. Our analysis shows that the increase of OC in New Delhi (and other cities in India) came from the increase of OC in residential sector, presumably due to the large share of biofuel uses in residential cooking that increased significantly during the lockdowns.

| Table 1: Relative change (%) of March-April average anthropogenic SO2, NH3, BC, and OC emissions         |        |                 |       |       |  |  |  |  |
|----------------------------------------------------------------------------------------------------------|--------|-----------------|-------|-------|--|--|--|--|
| averaged over 3°x3° box around the cities due to COVID lockdowns. Negative value indicates a decrease of |        |                 |       |       |  |  |  |  |
| emission due to the lockdowns.                                                                           |        |                 |       |       |  |  |  |  |
| City, Country                                                                                            | $SO_2$ | NH <sub>3</sub> | BC    | OC    |  |  |  |  |
| Shanghai, China                                                                                          | -16.1  | -11.7           | -14.4 | -10.4 |  |  |  |  |
| New Delhi, India                                                                                         | -28.2  | -12.3           | -2.1  | +1.5  |  |  |  |  |
| Los Angeles, USA                                                                                         | -26.3  | -16.4           | -27.7 | -19.2 |  |  |  |  |
| Paris, France                                                                                            | -34.3  | -6.8            | -27.0 | -5.3  |  |  |  |  |
| Lima, Peru                                                                                               | -45.9  | -11.1           | -28.9 | -10.5 |  |  |  |  |
| Kuwait City, Kuwait                                                                                      | -9.3   | -8.1            | -23.3 | -21.1 |  |  |  |  |

Figure 10 shows the observed and modeled PM<sub>2.5</sub> changes (2020-2019) as well as the GEOS attributions of bulk PM<sub>2.5</sub> changes into different aerosol components and into the lockdown and meteorological/natural effects. For brevity, we group the GEOS aerosol components into four broad groups, namely inorganic aerosols (including sulfate, ammonia, and nitrate), carbonaceous aerosols (including organic matter, black carbon, and brown carbon), dust, and sea-salt. Relative changes in total PM<sub>2.5</sub> between 2020 and 2019 derived from both the observations and GEOS simulations are listed in **Table 2**. To facilitate the discussion of potential effects of the differing meteorology and natural emissions on PM<sub>2.5</sub>, we also derived March-April averages of the planetary boundary layer height (PBLH), wind speed at 10 m, total precipitation rate, and OC emissions associated with fires from the GEOS simulation, as listed in **Table 3**. While PBLH affects PM2.5 through vertical mixing, wind speed at 10 m represents local ventilation conditions and affects dust emissions as well. The precipitation rate determines wet removals of aerosol. Major features for the individual stations are summarized in the following:

Shanghai, China: The observations show that PM<sub>2.5</sub> in 2020 was 19.5 μg m<sup>-3</sup> or 41.4% lower than that observed in 2019. In comparison, GEOS model shows a smaller reduction of PM<sub>2.5</sub> in 2020, i.e., 12.2 μg m<sup>-3</sup> or 18.4% (with respect to the modeled value), of which 7.5% is due to the lockdown-induced reduction in anthropogenic emissions and 11.8% due to the differing meteorology and natural emissions. Analysis of aerosol composition further shows that carbonaceous PM<sub>2.5</sub> made a larger contribution to the reduction of PM<sub>2.5</sub> than inorganic PM<sub>2.5</sub> did for both the lockdown-induced emissions reduction and the differing meteorology and natural emissions. In comparison to March-April 2019, PBLH and precipitation rate during the 2020

COVID-lockdowns increased by 90 m (18.2%) and 0.42 mm d<sup>-1</sup> (16.9%), respectively, both contributing to the decrease of surface PM<sub>2.5</sub>. Although fire-emissions increased by 83.3% from 2019 to 2020, the amount of fire emissions was relatively small and did not contribute significantly to PM<sub>2.5</sub>.

New Delhi, India: The observed PM<sub>2.5</sub> in 2020 was about 27 μg m<sup>-3</sup> or 36.2% smaller than that in 2019. GEOS model predicted a much smaller reduction of 4 μg m<sup>-3</sup> or 7.7% in 2020. The model also suggests that the differing meteorology (e.g., an increase of 115.5% in precipitation rate) and natural emissions (e.g., dust reduction associated with a 30.4% decrease of near-surface wind speed) constitutes of about the 4.6% reduction in PM<sub>2.5</sub>, while the lockdown-induced reduction in anthropogenic inorganic emissions for the 3.2% reduction.

**Los Angeles, USA**: Observed PM<sub>2.5</sub> in 2020 was lower than that in 2019 by 2.4 μg m<sup>-3</sup> or 18.4%. In comparison, the GEOS predicted a much smaller reduction of only 0.2 μg m<sup>-3</sup> or 3%. However, the 3% reduction in PM<sub>2.5</sub> is a balance of the 8.2% reduction in both inorganic and carbonaceous aerosols associated with the lockdown and the 5.7% increase in inorganic aerosol and sea salt due to differing meteorology. Dust and carbonaceous aerosol had much smaller increases, compared to inorganic aerosol and sea salt.

**Paris, France**: Although observed PM<sub>2.5</sub> decreased by 2.7 μg m<sup>-3</sup> or 17.5% from 2019 to 2020, GEOS model only predicted a reduction of 0.8 μg m<sup>-3</sup> or 8%. Further analysis of GEOS modeling experiments shows that the reduction of anthropogenic emissions due to the lockdowns contributed to a 5.1% reduction in PM<sub>2.5</sub>, which is evenly contributed by inorganic and carbonaceous PM<sub>2.5</sub>. Due to the differing meteorology between 2020 and 2019, dust PM<sub>2.5</sub> increased by 0.2 μg m<sup>-3</sup> while seasalt PM<sub>2.5</sub> decreased by 0.5 μg m<sup>-3</sup>. An increase of 0.3 μg m<sup>-3</sup> in carbonaceous PM<sub>2.5</sub> was associated with a 167.5% increase in wildfire emissions. However, differing meteorology reduced the formation of inorganic PM<sub>2.5</sub> by 0.3 μg m<sup>-3</sup> in 2020.

Lima, Peru: PM<sub>2.5</sub> was observed to decrease by 7.5 μg m<sup>-3</sup> or 36.3% in 2020. In comparison, the GEOS modeling predicted a relatively smaller reduction of 26.7% in PM<sub>2.5</sub>. Further analysis shows that the lockdown-induced anthropogenic emission reduction yielded a 16.7% reduction in PM<sub>2.5</sub>, with more reduction in inorganic than carbonaceous aerosol. The differing meteorology (e.g., about 10% increase in PBLH) reduced inorganic and carbonaceous PM<sub>2.5</sub> by a similar amount, which collectively contributed to the 12.7% reduction in total PM<sub>2.5</sub>.

**Kuwait** City, Kuwait: Different from the five cities discussed above, the observed reduction of 1.4 μg m<sup>-3</sup> in PM<sub>2.5</sub> in 2020 is smaller than the GEOS simulated reduction of 4.7 μg m<sup>-3</sup> by more than a factor of 3. However, due to GEOS model overestimated PM<sub>2.5</sub> in 2019 by more than a factor of 2, the relative reduction is 3.8% and 5.6% for the observation and GEOS model respectively. Furthermore, the GEOS model suggests that the reduction of 2.0% associated with the lockdowns is smaller than the 3.7% reduction due to the differing meteorology. The lockdown-induced reduction in inorganic PM<sub>2.5</sub> is more than a factor of two larger than that in carbonaceous PM<sub>2.5</sub>. One can also notice that dust PM<sub>2.5</sub> had a small reduction caused by the lockdown-induced anthropogenic emissions reduction, which might be attributed to the dynamic aerosol-radiation interactions accounted for in the GEOS simulations that affected both emissions and transport of dust. The differing meteorology had influenced PM<sub>2.5</sub> components differently. On one hand, the reduction of dust PM<sub>2.5</sub> by 6.5 μg m<sup>-3</sup> is consistent with the 56.3% decrease in near-surface wind speed. On the other hand, an increase of 2.6 μg m<sup>-3</sup> in inorganic PM<sub>2.5</sub> and an increase of 0.4 μg m<sup>-3</sup> in carbonaceous PM<sub>2.5</sub> are consistent with the 32.8% reduction in precipitation rate.

| Table 2: March-April average PM <sub>2.5</sub> in 2019 and relative change (%) of PM <sub>2.5</sub> between 2020 and 2019 derived from the observations and GEOS simulations. Negative values indicate that PM <sub>2.5</sub> was smaller in 2020 than 2019. The relative changes of 2020_COVID vs 2020_BAU and 2020_BAU vs 2019 represent contributions of the lockdown-induced anthropogenic emission reductions and the differing meteorology, respectively. |                                                             |                     |                            |                           |                               |                         |  |  |  |  |
|-----------------------------------------------------------------------------------------------------------------------------------------------------------------------------------------------------------------------------------------------------------------------------------------------------------------------------------------------------------------------------------------------------------------------------------------------------------------|-------------------------------------------------------------|---------------------|----------------------------|---------------------------|-------------------------------|-------------------------|--|--|--|--|
| City, Country                                                                                                                                                                                                                                                                                                                                                                                                                                                   | Observed PM <sub>2.5</sub> GEOS-simulated PM <sub>2.5</sub> |                     |                            |                           |                               |                         |  |  |  |  |
|                                                                                                                                                                                                                                                                                                                                                                                                                                                                 | 2019 (μg<br>m <sup>-3</sup> )                               | 2020 vs<br>2019 (%) | 2019 (μg m <sup>-3</sup> ) | 2020_COVID vs<br>2019 (%) | 2020_COVID vs<br>2020_BAU (%) | 2020_BAU vs<br>2019 (%) |  |  |  |  |
| Shanghai,<br>China                                                                                                                                                                                                                                                                                                                                                                                                                                              | 47.2                                                        | -41.4               | 66.3                       | -18.4                     | -7.5                          | -11.8                   |  |  |  |  |
| New Delhi,<br>India                                                                                                                                                                                                                                                                                                                                                                                                                                             | 73.6                                                        | -36.2               | 48.3                       | -7.7                      | -3.2                          | -4.6                    |  |  |  |  |
| Los Angeles,<br>USA                                                                                                                                                                                                                                                                                                                                                                                                                                             | 12.9                                                        | -18.4               | 7.2                        | -3.0                      | -8.2                          | +5.7                    |  |  |  |  |
| Paris, France                                                                                                                                                                                                                                                                                                                                                                                                                                                   | 15.5                                                        | -17.5               | 9.8                        | -8.0                      | -5.1                          | -3.0                    |  |  |  |  |
| Lima, Peru                                                                                                                                                                                                                                                                                                                                                                                                                                                      | 21.2                                                        | -36.3               | 15.5                       | -26.7                     | -16.7                         | -12.1                   |  |  |  |  |
| Kuwait City,<br>Kuwait                                                                                                                                                                                                                                                                                                                                                                                                                                          | 34.7                                                        | -3.8                | 83.3                       | -5.6                      | -2.0                          | -3.7                    |  |  |  |  |

| Table 3: Comparisons of some major meteorological variables and fire-emitted OC between 2020 and 2019   |              |                      |                                     |                         |  |  |  |  |
|---------------------------------------------------------------------------------------------------------|--------------|----------------------|-------------------------------------|-------------------------|--|--|--|--|
| on a basis of Mar-April average. The data are derived from GEOS simulations by averaging over 3°x3°     |              |                      |                                     |                         |  |  |  |  |
| box around the cities. Numbers in the parentheses represent the percentage change with respect to 2019. |              |                      |                                     |                         |  |  |  |  |
|                                                                                                         | PBLH (m)     | 10 m wind speed      | Precipitation (mm d <sup>-1</sup> ) | OC emissions from fires |  |  |  |  |
|                                                                                                         |              | (m s <sup>-1</sup> ) |                                     | $(g m^{-2} d^{-1})$     |  |  |  |  |
| Shanghai                                                                                                |              |                      |                                     |                         |  |  |  |  |
| 2019                                                                                                    | 493          | 1.19                 | 2.53                                | 5.3E-5                  |  |  |  |  |
| 2020                                                                                                    | 583          | 1.20                 | 2.95                                | 9.7E-5                  |  |  |  |  |
| 2020 vs 2019                                                                                            | +90 (+18.2%) | +0.01 (+0.8%)        | +0.42 (+16.9%)                      | +4.4E-5 (+83.3%)        |  |  |  |  |
| New Delhi                                                                                               |              |                      |                                     |                         |  |  |  |  |
| 2019                                                                                                    | 1073         | 1.59                 | 0.51                                | 1.8E-4                  |  |  |  |  |
| 2020                                                                                                    | 980          | 1.11                 | 1.09                                | 4.7E-5                  |  |  |  |  |
| 2020 vs 2019                                                                                            | -93 (-8.7%)  | -0.48 (-30.4%)       | +0.58 (+115.5%)                     | -1.3E-4 (-74.4%)        |  |  |  |  |
| Los Angeles                                                                                             |              |                      |                                     |                         |  |  |  |  |
| 2019                                                                                                    | 622          | 1.65                 | 0.65                                | 4.8E-5                  |  |  |  |  |
| 2020                                                                                                    | 633          | 1.48                 | 2.64                                | 1.7E-4                  |  |  |  |  |
| 2020 vs 2019                                                                                            | +11 (+1.8%)  | -0.17 (-10.1%)       | +1.99 (+304.3%)                     | +1.2E-4 (+247.5%)       |  |  |  |  |
| Paris                                                                                                   |              |                      |                                     |                         |  |  |  |  |
| 2019                                                                                                    | 728          | 1.94                 | 1.61                                | 4.1E-6                  |  |  |  |  |
| 2020                                                                                                    | 655          | 0.67                 | 1.78                                | 1.1E-5                  |  |  |  |  |
| 2020 vs 2019                                                                                            | -73 (-10.0%) | -1.27 (-65.7%)       | +0.17 (+10.6%)                      | +7.0E-6 (+167.5%)       |  |  |  |  |
| Lima                                                                                                    |              |                      |                                     |                         |  |  |  |  |
| 2019                                                                                                    | 322          | 1.91                 | 2.97                                | 1.7E-5                  |  |  |  |  |
| 2020                                                                                                    | 354          | 2.08                 | 2.43                                | 2.8E-5                  |  |  |  |  |
| 2020 vs 2019                                                                                            | +32 (9.9%)   | +0.17 (+8.7%)        | -0.54 (-18.1%)                      | +1.1E-5 (+63.2%)        |  |  |  |  |
| Kuwait City                                                                                             |              |                      |                                     |                         |  |  |  |  |
| 2019                                                                                                    | 732          | 1.41                 | 0.53                                | 2.1E-3                  |  |  |  |  |
| 2020                                                                                                    | 746          | 0.61                 | 0.36                                | 2.0E-3                  |  |  |  |  |
| 2020 vs 2019                                                                                            | +14 (1.9%)   | -0.80 (-56.3%)       | -0.17 (-32.8%)                      | -1.0E-4 (-5.3%)         |  |  |  |  |

Figure 10: Absolute differences (μg m<sup>-3</sup>) in March-April average PM<sub>2.5</sub> between 2020 and 2019 (a negative value indicating that PM<sub>2.5</sub> was smaller in 2020 than 2019) in Shanghai, China (a), New Delhi, India (b), Los Angeles, USA (c), Paris, France (d), Lima, Peru (e), and Kuwait City, Kuwait (f). Shown here include observed (yellow bars) and GEOS simulated (blue bars) changes in total PM<sub>2.5</sub>. The GEOS simulations are further classified into dust, inorganic aerosol, carbonaceous aerosol, and sea salt, shown below the total PM<sub>2.5</sub> change. For both total and component PM<sub>2.5</sub>,
 GEOS simulation is partitioned into GEOS PM<sub>2.5</sub> change due to anthropogenic emission change (orange) and GEOS PM<sub>2.5</sub> change due to differing meteorology (gray). Relative changes in PM<sub>2.5</sub> are listed in Table 2.

In summary, the GEOS modeling experiments show that both the lockdowns and the differing meteorology as well as natural emissions contribute to the PM<sub>2.5</sub> reduction in 2020, in comparison to that in 2019. In Shanghai, New Delhi, and Kuwait City, the differing meteorology and natural emissions made a larger contribution to the PM<sub>2.5</sub> reduction than the lockdowns did. On the other hand, the lockdown effects in Los Angeles, Paris, and Lima were larger than that due to the differing meteorology and natural emissions. Clearly, for all stations the effects of the differing meteorology and natural emissions need to be considered when interpreting and attributing the observed PM<sub>2.5</sub> reduction in 2020 to the lockdown induced reductions in anthropogenic emissions. We also want to point out that large differences exist between the observed and modeled PM<sub>2.5</sub> for most of these stations. Although it is difficult to pinpoint these large discrepancies in a quantitative way, sources of errors would include several aspects of uncertainty in GEOS modeling. First, the relative contributions to emissions from different sources or sectors in CEDS may have large uncertainties (Hoesly et al., 2018). Natural emissions from wildfires and dust would also be very uncertain. As documented in a recent paper (Collow et al., 2024), GEOS simulated aerosol components have large discrepancies against surface observations. Second, the sector-dependent adjusting factors for COVID-lockdowns based on the mobility data may be subjected to large uncertainties due to assumptions of relationships between anthropogenic emissions and mobility (Forster et al., 2020). Third, GEOS modeling of meteorological effects on PM<sub>2.5</sub> concentration may be also biased, due to uncertainties associated with meteorological fields themselves and/or parameterizations of aerosol removal processes.

# 3.3. Observed changes in PM<sub>2.5</sub> during the major lockdown period in other stations.

In previous sections, we have presented a detailed analysis in the six representative stations about the responses of surface PM<sub>2.5</sub> to the lockdowns marked by the stringency index as well as to the meteorological conditions and natural aerosol events. Although similar analysis has been performed for all the stations, for the sake of brevity here we only present monthly PM<sub>2.5</sub> during and prior to the pandemic years for other 16 stations in 15 cities (Jakarta South and Central sites are presented separately), as shown in **Figures 11** and **12.** When examining these plots along with the stringency index of lockdowns (Figures S1 and S2), we observe that eight stations in Figure 11 show the reduced monthly PM<sub>2.5</sub> in correspondence to the elevation of stringency index. These stations are located in China, India, and United States. For those in China, the high stringency index was recorded not only in early 2020 (February – April) but also in March-October of 2022. On the contrary,

the stations in Figure 12 generally do not show a clear decrease in monthly PM<sub>2.5</sub> corresponding to the lockdowns. When further looking into the source attributions of PM<sub>2.5</sub> in individual stations (Figure 2), we notice that those stations showing the reduced PM<sub>2.5</sub> in correspondence to the lockdowns generally have a higher contribution (e.g., >47%) from the LERS, a much smaller fraction (16-27%) for residential sources, and no significant contribution (<6%) from episodic events such as dust storms. On the contrary, those stations without displaying a decrease in PM<sub>2.5</sub> corresponding to the high stringency index have a lower LERS contribution (e.g., <40%) and a relatively higher contribution (20-50%) by the residential sector (Jakarta and Hanoi, in particular), or the predominance of desert dust (e.g., Dubai, Manama, and Addis Ababa). It is necessary to note that extremely high PM<sub>2.5</sub> in spring and summer of 2022 in Dubai may suggest a potential problem associated with the measurements in this station. Although high dust events occurred frequently in the region in 2022 (Francis et al., 2023), Manama and Kuwait City in the region did not record as high PM<sub>2.5</sub> as that in Dubai. Further analysis of AERONET monthly AOD in Dubai (see Figure S9 in the supplement) shows that AOD in May 2022 had a similar magnitude to that in May 2019, contrary to the large difference in the observed surface PM<sub>2.5</sub>.

In summary, our analysis shows that there were no systematic reductions in PM<sub>2.5</sub> in response to the reduced human mobility due to the implementation of lockdown measures. In some urban areas, the coincidences of decreasing PM<sub>2.5</sub> and increasing stringency index strongly suggest the impact of COVID-lockdowns on improving air quality. In fact, the lockdowns yielded a historic low PM<sub>2.5</sub> level in several cities. In other urban areas, there was no reduction of PM<sub>2.5</sub> in response to the lockdown, suggesting that the impact of lockdowns could have been compounded with other factors such as meteorological conditions and natural emissions (desert dust and wildfire smoke) to some extent. Our analysis also manifests the importance of PM<sub>2.5</sub> source attributions in determining how the level of PM<sub>2.5</sub> responds to the lockdowns. When the energy, industry, transportation, commercial and other combustions, AFCID (anthropogenic fugitive, combustion, and industrial dust), and international shipping sectors are major attributors of PM<sub>2.5</sub>, it would be easier to detect the PM<sub>2.5</sub> reduction in response to the lockdowns. On the other hand, when residential sector and/or natural emissions such as dust or wildfires are major contributors of PM<sub>2.5</sub>, the response of PM<sub>2.5</sub> to the lockdowns might be masked out and even an increase of PM<sub>2.5</sub> could occur during the lockdown periods. Finally, it is important to bear in mind that even for those cities with the lowest level of PM<sub>2.5</sub> in the recent decade occurred during the COVID-19 lockdowns, the observed PM<sub>2.5</sub> reductions should not be attributed fully to the lockdowns. The variability in meteorological conditions and natural emissions might have made sizable contributions.

Figure 11: Monthly variations of PM<sub>2.5</sub> concentration in 2019 (brown), 2020 (red), 2021 (orange), 2022 (blue), and the pre-pandemic climatology (with black line denoting the average and gray shaded areas the range of observations during the pre-pandemic period) in Beijing (a), Shenyang (b), Guangzhou (c), Chennai (d), Hyderabad (e), Kolkata (f), Mumbai (g), and New York City (h). The time series of stringency index for these cities are shown in Figure S1.

Figure 12: Monthly variations of PM<sub>2.5</sub> concentration in 2019 (brown), 2020 (red), 2021 (orange), 2022 (blue), and the pre-pandemic climatology (with black line denoting the average and gray shaded areas the range of observations during the pre-pandemic period) in Milano (a), Pristina (b), Hanoi (c), Jakarta (central and south) (d and e), Addis Ababa (f), Dubai (g), and Manama (h). The time series of stringency index for these cities are shown in Figure S2.

## 4. Discussion

## 4.1. Can satellite observations of AOD be used to detect surface PM2.5 changes in response to the lockdowns?

The ground-based observations of PM<sub>2.5</sub> are inherently limited in spatial coverage. Numerous studies have explored the use of satellite observations of AOD to estimate the surface PM<sub>2.5</sub> that can fill the spatial gaps of surface based PM<sub>2.5</sub> measurements (van Donkelaar et al., 2010; Wei et al., 2021). Satellite remote sensing is appealing in this regard because it provides routine observations of AOD on a global scale and over a multi-year or even multi-decade time scale. Can such satellite remote sensing observations be used to discern the lockdowns' impacts on surface PM<sub>2.5</sub>? To answer this question, we sample MODIS/Aqua AOD over the 21 cities in 2019 (a pre-pandemic year) and 2020 (the pandemic year). We then calculate the relative changes of March-April average AOD values between 2020 and 2019 as follows:

$$AOD\ change\ (\%) = \frac{AOD_{2020} - AOD_{2019}}{AOD_{2019}} \times 100\%$$

The relative changes of observed March-April average PM<sub>2.5</sub> at the location of the same sites are calculated with similar methodology. From the modeling perspective, we compute the relative changes of AOD and PM<sub>2.5</sub> based on GEOS simulations for 2019 and 2020 (for the COVID scenario, namely 2020-COVID, as discussed earlier).

Figure 13 compares the relative changes of AOD and PM<sub>2.5</sub> in March-April from 2019 to 2020 over the 21 cites from both observational (left panel) and modeling (right panel). The observations show that in half (10) of those 20 cities (excluding Hanoi because of the lack of PM<sub>2.5</sub> data in March and April of 2019) where the changes of AOD and PM<sub>2.5</sub> are in the same directions. In the other half the changes are opposite. On the other hand, the GEOS model shows a majority (17) of the 20 cities having the same directions of AOD and PM<sub>2.5</sub> changes. Quantitatively, AOD and PM<sub>2.5</sub> changes can differ substantially for both observations and modeling, depending on cities. This presumably reflects the complexity of local/regional aerosol sources including both anthropogenic and natural, transported amount, and their dependance on altitudes. Our analysis suggests that satellite AOD, despite its great advantage in spatial and temporal sampling, cannot always be used to detect changes in surface PM<sub>2.5</sub> quantitatively (e.g., percentage of change) or even qualitatively (e.g., direction of change). Previous studies using satellite AOD measurements for detecting or inferring the COVID-lockdown's impacts on PM<sub>2.5</sub> air quality may need to be reassessed. There is also a possibility that MODIS AOD is subject to large uncertainties (e.g.,15-20% at least), particularly in urban areas where the surface is bright and highly heterogeneous (Levy et al., 2013). It is understandable that when the AOD

uncertainty in satellite remote sensing is comparable to or larger than the interannual variability, the value of using satellite remote sensing product for discerning the changes is greatly reduced.

Figure 13: Relative changes (%) of March-April average PM<sub>2.5</sub> and AOD between 2020 and 2019 in 20 stations based on the observations (a) and GEOS simulations (b).

## 615 4.2. Discrepancies in PM<sub>2.5</sub> changes between observations and GEOS simulations

To use the GEOS modeling for attributing the observed changes in PM<sub>2.5</sub> to different factors, it requires that the GEOS simulations and observational data are at least consistent in the direction of PM<sub>2.5</sub> changes. This fundamental requirement was met for the six stations discussed in section 3.2.2, albeit the magnitude of the changes in the GEOS results was smaller than observed. Here we examine observation-model discrepancies in the PM<sub>2.5</sub> changes for 20 stations, as shown in **Figure 14**, both for absolute (left panel) and relative (right panel) changes in the March-April average PM<sub>2.5</sub> concentration (i.e., 2020 minus 2019). Note again that the Hanoi/Vietnam station was excluded because it did not have PM<sub>2.5</sub> observations in March-April 2019. We found that 17 out of 20 stations show consistent directional changes in PM<sub>2.5</sub> concentration between observations

and GEOS simulations. Of these, 13 stations had negative differences (2020 lower than 2019) and four stations had positive differences (2020 higher than 2019). Opposite changes in PM<sub>2.5</sub> between the observations and GEOS modeling were found in the remaining three stations, namely Guangzhou/China, Chennai/India, Jakarta/Indonesia. Quantitatively, the observation-model disparity can be substantial, depending on stations. For all the stations the relative changes in the observed PM<sub>2.5</sub> between 2020 and 2019 range from -41.4% to +21.0%, while the GEOS modeling gives a relatively smaller range of -25.2% to +10.6%.

The opposite changes in PM<sub>2.5</sub> between the observation and GEOS modeling warrant further analysis of the GEOS modeling experiments. **Figure 15** shows the GEOS attributions of simulated bulk PM<sub>2.5</sub> changes into different aerosol components and into the lockdown and meteorological/natural effects for Guangzhou in China and Jakarta in Indonesia. Chennai in India has a pattern of change similar to Jakarta but with a much smaller magnitude, which is shown along with other 11 stations in the supplement (**Figure S10**). We also include changes in major meteorological variables and OC emissions associated with wildfires in **Table S2** of the supplement for all the stations not included in Table 3. In Guangzhou, the observation shows a decrease of 1.3 μg m<sup>-3</sup> in PM<sub>2.5</sub>. However, GEOS simulations suggest that the effect of anthropogenic emission reduction associated with the lockdown is overcompensated by an increase of PM<sub>2.5</sub> due to the differing meteorology (e.g., a 38% decrease of precipitation rate, see Table S2) and natural emissions (e.g., a 215% increase of wildfire emission, see Table S2), leading to an overall increase of 5.7 μg m<sup>-3</sup> in PM<sub>2.5</sub>. It is possible that the precipitation rate was too low, and the wildfire emissions were too high in 2020 in the GEOS modeling. An opposite pattern of changes is displayed in Jakarta. While the observation shows an increase of 6.0 μg m<sup>-3</sup> in PM<sub>2.5</sub>, GEOS modeling gives a decrease of 6.6 μg m<sup>-3</sup> in PM<sub>2.5</sub> – a result of the reduced anthropogenic emissions and the differing meteorology/natural emissions (e.g., a 30.9% of increase in precipitation rate and a 12.6% of decrease in wildfire emissions, see Table S2). It is possible that sources of PM<sub>2.5</sub> used in GEOS simulations may have significant biases in Jakarta.

To better explain the observed PM<sub>2.5</sub> changes with the model simulations, future research is clearly needed to improve the model's performance. One of such endeavors is to improve the accuracy of emission inventories, particularly the relative importance of individual source sectors. The estimated emission reduction used in the GEOS model simulation was obtained by applying the adjustment factors to the 2019 emission based on preliminary, incomplete information from the Apple/Google

mobility data, which could have large uncertainties. The newly released CEDS emission that covers the emission inventory during the COVID years is expected to be more accurate for use in the model for future studies.

Figure 14: Changes in PM<sub>2.5</sub> concentration (March-April average, 2020 minus 2019) from the observations (orange bars) and GEOS simulations (blue bars) in 20 stations. Left panel (a) is for absolute change in PM<sub>2.5</sub> ( $\mu$ g m<sup>-3</sup>) and right panel (b) for relative change (%).

Figure 15: Absolute differences (μg m<sup>-3</sup>) in March-April average PM<sub>2.5</sub> between 2020 and 2019 (a negative value indicating that PM<sub>2.5</sub> was smaller in 2020 than 2019) in Guangzhou China (a), and Jakarta, Indonesia (b). Shown here include observed (yellow bars) and GEOS simulated (blue bars) changes in total and component PM<sub>2.5</sub>. For both total and component PM<sub>2.5</sub>, GEOS simulation is partitioned into GEOS PM<sub>2.5</sub> change due to anthropogenic emission change (orange) and GEOS PM<sub>2.5</sub> change due to differing meteorology (gray).

#### 5. Conclusion

Our analysis of multi-year surface PM<sub>2.5</sub> observations in 21 cities around the globe shows that reductions in PM<sub>2.5</sub> (particulate matter with an aerodynamic diameter of less than 2.5 µm) did not occur systematically in all the cities in response to the reduced human activities due to the implementation of COVID-19 lockdown measures. In some cities, the decreasing PM<sub>2.5</sub> was coincident with the increasing stringency index and yielded a historic record of low PM<sub>2.5</sub> level, strongly suggesting the

impact of COVID-lockdowns on improving the air quality. On the contrary, in other cities there was no reduction of PM<sub>2.5</sub> in response to the lockdown, suggesting that the positive impact of lockdown-induced emission reductions on air quality could have been compounded with other factors such as meteorological conditions and emissions/transport of desert dust and wildfire smoke. Our analysis also suggests that the PM<sub>2.5</sub> source attributions determine how the level of PM<sub>2.5</sub> responds to the lockdowns. When the potential lockdown emission reduction sectors (LERS) such as the energy, industry, transportation, combustions, anthropogenic fugitive, combustion, and industrial dust (AFCID), and shipping sectors predominates over the residential sector, it was easier to detect the PM<sub>2.5</sub> reduction in response to the lockdowns. On the other hand, when residential sector and/or natural emissions such as dust storms and wildfires were more important contributors of PM<sub>2.5</sub> than those potential LERS, the response of PM<sub>2.5</sub> to the lockdown-induced anthropogenic emission reductions might be masked out and even an increase of PM<sub>2.5</sub> could occur during the lockdown. This manifests the importance of PM<sub>2.5</sub> source attributions in developing effective pollution control strategies for improving the air quality. Results of this study provide a preview of potential mixed effects on urban air quality when transitioning gasoline and diesel-powered vehicles to electric vehicles.

Even for those cities with the lowest level of PM<sub>2.5</sub> in the recent decade that were coincident with the elevation of stringency index, the observed PM<sub>2.5</sub> reductions should not be attributed fully to the lockdowns. The non-lockdown effect resulting from variabilities in meteorological conditions and natural emissions might have made sizable contributions. The analysis of Goddard Earth Observing System (GEOS) modeling experiments suggests that effects other than anthropogenic emissions were significant and could well exceed the lockdown effect in some cases.

Our analysis also suggests that the GEOS model is still subject to large uncertainties and may not be used to reliably attribute the observed PM<sub>2.5</sub> changes to changes in anthropogenic emissions, natural emissions, and meteorological conditions. The model and observations show disparities in the quantitative change in PM<sub>2.5</sub>. In some cases, the direction of change in PM<sub>2.5</sub> can be opposite between the model and observations, which makes it impossible to use the model to interpret the observations. This manifests the importance of continuous efforts on improving modeling performance in general and the source apportionments of PM<sub>2.5</sub> particularly. The change of anthropogenic emissions in each sector during COVID is subject to large uncertainties, and the model results shown here should be considered more as a "sensitivity" study. When the gridded Community Emission Data System (CEDS) 2024 release is prepared, we could rerun the model for a more systematic

assessment. Despite the advantage of satellite remote sensing in terms of routine daily sampling over decadal time spans, using aerosol optical depth (AOD) observations from satellites cannot always detect the impacts of COVID-19 lockdowns on the PM<sub>2.5</sub> air quality and even the columnar aerosol loading. This is a result of the complex and non-proportional relationship between AOD and surface PM<sub>2.5</sub>, as well as the large uncertainty in AOD retrievals from the Moderate Resolution Imaging Spectrometer (MODIS).

## **Author contribution**

MC designed research. QT acquired hourly PM<sub>2.5</sub> data and processed daily average PM<sub>2.5</sub>. CY carried out most of data analysis with the PM<sub>2.5</sub> observations. HB and PC ran the GEOS model experiments. HY and HB analyzed the GEOS model outputs. HY, CY and MC wrote the initial draft of this paper. PC, QT, and HB helped review and edit the paper. All authors made substantial contributions to this work.

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
