# Peer review of "Assessing COVID-19 Lockdowns' Impacts on Global Urban PM2.5 Air Quality with Observations and Modeling"

_EGUsphere, 2025_

## Author Comment (AC3)

**Reviewer#2**

Having reviewed the paper, I find it to be scientifically sound and well-structured. However, I would recommend the following minor revisions to enhance its clarity and impact:

The discrepancies between GEOS model predictions and observations in cities like Guangzhou and Jakarta warrant further explanation. Perhaps including a brief analysis of the local emission inventory uncertainties would strengthen this section.

*Response:* Thanks for the excellent suggestion. We further analyzed our GEOS simulations (e.g., changes in PM2.5 components due to anthropogenic emission reductions and differing meteorology and natural emissions between 2020 and 2019) to better understand potential uncertainties contributing to the model's deficiency in capturing the observed PM2.5 changes in Guangzhou and Jakarta. The following figure (Figure 15) will be added in the revised paper with a discussion of modeling uncertainties. Similar charts for all other cities will also be provided in the supplementary material (Figure S10).

[Figure]

[Figure]

**Figure 15:** Absolute differences ($\mu g\ m^{-3}$) in March-April average $PM_{2.5}$ between 2020 and 2019 (a negative value indicating that $PM_{2.5}$ was smaller in 2020 than 2019) in Guangzhou, China (a),

and Jakarta, Indonesia (b). Shown here include observed (yellow bars) and GEOS simulated (blue bars) changes in total $PM_{2.5}$. The GEOS simulations are further classified into dust, inorganic aerosol, carbonaceous aerosol, and sea salt, shown below the total $PM_{2.5}$ change. For both total and component $PM_{2.5}$, GEOS simulation is partitioned into GEOS $PM_{2.5}$ change due to anthropogenic emission change (orange) and GEOS $PM_{2.5}$ change due to differing meteorology (gray).

The meteorological analysis could benefit from additional discussion of how specific weather patterns during the lockdown periods might have influenced $PM_{2.5}$ concentrations independently of emission changes.

**_Response:_** Thanks for the suggestion. We will analyze several meteorological variables including boundary layer height, winds, precipitation, as well as natural emissions (wildfires, dust, and sea-salt) modulated by meteorology to understand how the weather patterns during the lockdown periods might have influenced $PM_{2.5}$ concentrations.

Review terminology throughout the paper.

**_Response:_** We will review carefully to make sure a consistent and appropriate use of terminology. Thanks.

Overall, this is a valuable contribution that effectively leverages the "natural experiment" of COVID-19 lockdowns.

---

## Author Response (AR1)

**Dear Editor:**

We very much appreciate insightful comments on our manuscript by two reviewers and have carefully considered all the comments and addressed them in this revision. We believe that the quality of this revised paper has been significantly improved.

Below is our point-by-point response to the reviewers' comments. We are also including the revised paper with track-changes so that you can easily gather how we have addressed the comments and suggestions.

10 We look forward to your further comments on the revised paper.

Sincerely,

Hongbin Yu
on behalf of all the co-authors

**Reviewer #1**

**20 General comments:**

In the paper, "Assessing COVID-19 Lockdowns' Impacts on Global Urban PM2.5 Air Quality with Observations and Modeling", the authors investigated surface PM2.5 concentrations in 21 cities across the globe during the COVID-19 lockdowns and pre-pandemic years through the use of *in situ*, remotely sensed, and model data. The authors conducted a thorough study of an important topic, and the current manuscript is generally of high quality and presentation. However, I do have some concerns and corrections (as outlined below) that I believe need to be addressed before the manuscript is suitable for publication in ACP. Overall, I recommend minor revisions for this paper.

**Specific comments:**

-As for a general comment concerning grammatical errors, these are found throughout the manuscript. I tried to capture several of these (as found in the technical corrections comments below), but the manuscript likely needs another round of thorough proofreading and technical editing.

**Response:** We are grateful to the reviewer for capturing grammatical errors in the manuscript. We will do technical editing when address review comments and attempt a thorough proofreading before submitting the revised paper.

-Abstract: Define all acronyms, such as  $PM_{2.5}$ , GEOS, etc.

**Response:** Yes, we did.**

-Page 4, Fig. 1: Add units label to the color bar. Also, why is Greenland and Antarctica in white on the map? Noting this because the far north polar region above Greenland is colored in blue. The reason for this should be explained in the caption or text of the paper (or modified in the figure).

Response: The white Greenland and Antarctica in original map was due to that the PM2.5 concentration is less than 1 μg m-3 in these areas, which is beyond the original lower bound of the color bar. We remade the figure by adding units label to the color bar, changing the lower bound of the color bar from 1 to 0, and adding names corresponding to the 21 cities underneath the map.

-Page 5, Line 123: Define all acronyms in the parentheses.

**Response: done.**

-Page 6, Lines 153-154: Why 5-day running means and not 3 or 7? Were sensitivity studies conducted to determine the temporal length used here?

**Response:** The objective of applying 5-d moving average is to remove high-frequency variation of PM2.5 due to the control of synoptic conditions so that potential signals of the lockdowns could be easily detected. The selection of 5-d, instead of 3-d or 7-d, moving average is a compromise of detecting the lockdown signal and keeping the synoptic-scale variations of PM2.5. It would not affect major conclusions of the study. Below is an example that compares 3-d, 5-d, and 7-d running means in Shanghai.

-Figure 9: I'm concerned about the robustness of the regression lines for so few points. This is especially true for Fig. 9f (Kuwait City), for which there are only 3 pre-pandemic points. Related to this, on Page 21, Lines 384-385, the corresponding R2 value of 0.907 is noted as a reason for a statistically significant trend. But again, this is only for 3 points. Can you please comment on this?

**Response:** We agree that the limited data points in the  $PM_{2.5}$  observations (i.e., Lima and Kuwait City) make the  $R^2$  value less meaningful in terms of statistical significance. For Shanghai and Paris, the pre-pandemic trend is statistically significant with p = 0.01 based on the student t-test. We have clarify these points in the revised manuscript.

-Page 24, Lines 427-433: Add some discussion/more details on how the numbers in Table 1 were arrived at?

**Response:** We derived the numbers in Table 1 by calculating differences of March-April emissions around the six cities (averaged over a 3°x3°box around each city) between 2020-COVID and 2020-BAU scenarios of anthropogenic emissions that were used to drive the GEOS simulations. As described in section 2.2, for the 2020-BAU scenario, anthropogenic emissions in 2019 were used to represent the baseline emissions of 2020, assuming the anthropogenic emissions would not have significant changes from 2019 to 2020 in a business-as-usual scenario. For the 2020-COVID scenario, the 2019 anthropogenic emissions in individual sectors were adjusted (decreased or increased, depending on sectors) based on daily mobility data gathered by Apple and Google to reflect the COVID lockdown's impacts on anthropogenic emissions, which was developed by Foster et al. (2020). We have provided the details in the revised manuscript.

-Page 28, Lines 489-495: Concerning the analysis discussed here (corresponding to Table 2 and Figure 10), please add some discussion/details on the differences between the observed  $PM_{2.5}$  and modeled  $PM_{2.5}$ . For example, in New Delhi, there is a large discrepancy between these two ( $\sim$  -36% for observations vs.  $\sim$  -8% for modeled). I don't believe the current narrative is sufficient in explaining this discrepancy.

**Response:** We believe that the large differences between the observations and GEOS simulations could have come from three sources associated with the GEOS modeling, although it is difficult to quantify these errors. First, the relative contributions to emissions from different sources or sectors in CEDS may have large uncertainties (Hoesly et al., 2018). As documented in a recent paper (Collow et al., 2024), GEOS simulated aerosol components have large discrepancies against surface observations. Second, the sector-dependent adjusting factors based on the mobility data may be subjected to large uncertainties due to assumptions of relationships between anthropogenic emissions and mobility (Forster et al., 2020). Third, GESO modeling of meteorological effects on PM2.5 concentration may be also biased, due to uncertainties associated with meteorological fields themselves and/or parameterizations of aerosol removal processes.

Collow, A. B., et al., Benchmarking GOCART-2G in the Goddard Earth Observing System (GEOS), Geoosci. Model Dev., 17, 1443-1468, 2024.

Forster, P. M., et al., Current and future global climate impacts resulting from COVID-19. Nature Climate Change, 10, 913-919, 2020.

Hoesly, R. M., et al., Historical (1750–2014) anthropogenic emissions of reactive gases and aerosols from the Community Emissions Data System (CEDS), Geosci. Model Dev., 11, 369–408, 2018.

-Page 29, Lines 511-514: Were these high PM2.5 observations looked into further? What about looking at any ground-based AERONET AODs in the region to confirm the possible dust events?

**Response:** Thanks for the suggestion. We analyzed monthly AOD at 500 nm from an AERONET station in Dubai (DEWA\_Research\_Centre). As shown in figure below. The AOD in May 2022 had a similar magnitude to that in May 2019. On the other hand,  $PM_{2.5}$  in May 2022 was more than 4 times that in May 2019 (Fig. 12g). This large discrepancy likely suggests a problem in the  $PM_{2.5}$  observations. We modified the text and included the figure below in the supplemental material.

-Page 32, Lines 573-575: "Previous studies using satellite AOD measurements for detecting or inferring the COVID-lockdown's impacts on PM2.5 air quality need to be reassessed." In my opinion, this is a strong statement to use. Are you referring to specific studies here, particularly those that came to opposite conclusions of this paper?

**Response:** We were not referring to specific studies here. We just wanted to caution that the use of AOD change between 2020 and pre-pandemic years may not tell us how PM2.5 has changed in terms of either magnitude (in percentage) or even the direction. We have rephrased it to make it a less strong statement.

-In the Conclusion section, redefine all acronyms.

**Response: done.**

-Page 35, Line 630: About how large are these uncertainties, and how might that impact the results of this study?

**Response:** The uncertainties could be as large as a factor of 4 (e.g., New Delhi), which made it impossible for attributing the observed changes in  $PM_{2.5}$  to changes in emissions and meteorology quantitatively. We have revised this paragraph to make the message delivered more clearly.

**Technical corrections:**

- 125 **Response:** We appreciate the reviewer's careful reading of the paper and suggestions for technical corrections. In this revised paper, we have corrected all the technical errors listed below and some additional errors we found during the revision.
  - -Page 2, Line 48: Change "the extended period" to "an extended period"
- -Page 6, Fig. 2: center the title "Sector Contributions (%)". Also, increase the text of the color labels below the chart (as well as the left y-axis city labels, if there is room to do so).
  - -Page 7, Line 158: Add "the" before "Aqua"
  - -Page 7, Line 177: Add a comma after "salt"
  - -Page 8, Line 197: Add "to" after "refer"
  - -Page 8, 198: Add "is referred to" after "2019"
- 135 -Page 8, Line 200: Add "the" before "observed"
  - -Page 9, Line 213: Add "the" before "other"

- -Page 9, Line 226: Suggest replacing "in selected stations" with "of selected stations"
- -Page 10, Line 227: Replace "in the six stations" with "at the six stations"
- -Page 10, Line 232: Replace "climatology" with "climatologies"
- -Page 10, Line 235: A word seems to be missing after "following"....did you mean to state "following sections" or "following subsections"?
  - -Figures 3 through 8, in all labels/titles:
  - -Replace "PM2.5" with the subscript for 2.5, such as "PM2.5"
  - -Replace "ug/m3" with " µg m3"
- 145 -Page 10, Line 244: Add a period to the end of the sentence before "Similarly"
  - -Page 11, Lines 260-261: I suggest keeping this sentence as part of the previous paragraph rather than keep it as its own paragraph.
  - -Page 12, Lines 291-293: As the previous comments, I suggest moving this sentence to the end of the previous paragraph.
- 150 -Page 12, Line 297: Replace "the record" with "a record"
  - -Page 12, Line 301: Replace "essay" with "paper"
  - -Page 21, Line 376: Add "the" after "if"
  - -Figure 9: Edits to the figure are needed here, including enlarging the text of the axis labels and tick marks. Also, in all labels/titles:
- $^{155}$  -Replace "PM2.5" with the subscript for 2.5, such as "PM $_{2.5}$ "
  - -Replace "ug/m^3" with " µg m-3"
  - -Page 24, Line 426: Add "the" before "aerosol"

- -Figure 10: Replace "PM2.5" with the subscript for 2.5, such as "PM2.5". Also, add the units of PM2.5 somewhere in the figure itself, not just the caption.
- 160 -Page 28, Line 496: Remove the period after "cities"
  - -Page 28, Line 501: Change "11-12" to "11 and 12".
  - -For Figures 11 and 12, add labels (a) through (h). Also, in all labels/titles:
  - -Replace "PM2.5" with the subscript for 2.5, such as "PM2.5"
  - -Replace "ug/m^3" with " µg m-3"
- -Figure 13: Add labels (a) and (b). Also, replace "PM2.5" with the subscript for 2.5, such as "PM $_{2.5}$ ", and center the titles of each plot.
  - -Page 33, Line 584: Remove the period after "simulations"
  - -Figure 14: Add labels (a) and (b). Also, replace "PM2.5" with the subscript for 2.5, such as "PM $_{2.5}$ ", and center the titles of each plot. Replace "ug/m $^3$ " with " µg m $^3$ ".
- 170 -Page 35, Line 623: "provides" should be "provide".
  - -Page 35, Line 625: Please check the grammar in this sentence, as "was" might need to be changed to "were".
  - -Page 35, Line 632: "observation" should be "observations" and add "the" before "quantitative"
- -Page 36, Lines 638-641: These two sentences should be moved to the end of the previous paragraph.
  - -Page 36, Line 641: Remove "and". Also, I suggest replacing "measurements" with "retrievals".

**Reviewer#2**

- Having reviewed the paper, I find it to be scientifically sound and well-structured. However, I would recommend the following minor revisions to enhance its clarity and impact:
  - The discrepancies between GEOS model predictions and observations in cities like Guangzhou and Jakarta warrant further explanation. Perhaps including a brief analysis of the local emission inventory uncertainties would strengthen this section.
- 185 Response: Thanks for the excellent suggestion. We further analyzed our GEOS simulations (e.g., changes in PM2.5 components due to anthropogenic emission reductions and differing meteorology and natural emissions between 2020 and 2019) to better understand potential uncertainties contributing to the model's deficiency in capturing the observed PM2.5 changes in Guangzhou and Jakarta. The following figure (Figure 15) is added in the revised paper with a discussion of modeling uncertainties.
   190 Similar charts for all other cities are also provided in the supplement (Figure S10).

**Figure 15:** Absolute differences (μg m-3) in March-April average PM2.5 between 2020 and 2019 (a negative value indicating that PM2.5 was smaller in 2020 than 2019) in Guangzhou, China (a), and Jakarta, Indonesia (b). Shown here include observed (yellow bars) and GEOS simulated (blue bars) changes in total PM2.5. The GEOS simulations are further classified into dust, inorganic aerosol, carbonaceous aerosol, and sea salt, shown below the total PM2.5 change. For both total and component PM2.5, GEOS simulation is partitioned into GEOS PM2.5 change due to anthropogenic emission change (orange) and GEOS PM2.5 change due to differing meteorology (gray).

200

195

The meteorological analysis could benefit from additional discussion of how specific weather patterns during the lockdown periods might have influenced PM2.5 concentrations independently of emission changes.

205 Response: Thanks for the suggestion. We analyzed several meteorological variables including boundary layer height, winds, precipitation, as well as natural emissions (wildfires, dust, and sea-salt) modulated by meteorology to understand how the weather patterns during the lockdown periods might have influenced PM2.5 concentrations. Table 3 and Table S2 have been included and discussed in this revision.

210

Review terminology throughout the paper.

**Response:** We reviewed the revised paper carefully to make sure a consistent and appropriate use of terminology. Thanks.

Overall, this is a valuable contribution that effectively leverages the "natural experiment" of COVID-19 lockdowns.